# Government Emergency Response Assessment from a Novel Perspective of GB/T37228-2018—A Case Study in China

**DOI:** 10.3390/ijerph20065091

**Published:** 2023-03-14

**Authors:** Jiangdong Bao, Yu Bao

**Affiliations:** 1School of Economics and Management, Hanshan Normal University, Chaozhou 521041, China; 2School of Mathematics and Statistics, Henan University, Kaifeng 475001, China

**Keywords:** government emergency response, two-tuple linguistic information, GB/T37228-2018, H Government

## Abstract

With the development of the economy and science and technology, the threat of various emergencies has brought severe governance challenges to governments. In order to minimize the harm and loss of emergencies and further improve the authority and credibility of government, this study uses the two-tuple linguistic information method to assess the indicator system of H Government of China constructed according to the GB/T37228-2018 standard (Societal security—Emergency management—Requirements). The result shows that the management of emergency resource management, information collection methods, response and guarantee plans, and other aspects is relatively standardized. However, the middle and late stages of emergency management are relatively weak, which is mainly reflected in the continuity of situation assessment, in information sharing and feedback, and in the coordination process. The current work implies that the GB/T37228-2018 standard broadens the approach of government emergency response assessment and strengthens the standardization process of emergency response. It also challenges the implicit knowledge of emergency response, the integration of time and space variables, and other issues.

## 1. Introduction

The rapid development of science and technology has promoted an exponential surge in the productivity in modern society. At the same time, the quantity of potential dangers and threats has reached a new height. At present, countries around the world are faced with enormous opportunities and challenges for development, change, and adjustment. Increasing instability, uncertainty factors, and the continuing spread of non-traditional security threats have brought severe challenges to governments. Public health emergencies, public security incidents, natural disasters, accidents and disasters, and other crisis events occur from time to time. The effects of these events seriously threaten people’s physical and mental health and the safety of their property, affect the smooth operation of normal social order, and even endanger national security. How to enhance the capacity for disaster prevention, mitigation, and disaster resistance and relief and optimize the management of various public emergencies is an important issue that the government, as the core of the public sector, must seriously consider. Internationally, the United Nations Office for Disaster Risk Reduction defines an Early Warning Systems (EWS) for natural hazards as ‘an integrated system of hazard monitoring, forecasting and prediction, disaster risk assessment, communication and preparedness activities systems and processes that enables individuals, communities, governments, businesses and others to take timely action to reduce disaster risks in advance of hazardous events’ [1]. EWSs have four components: risk knowledge, monitoring and forecasting, dissemination and communication, and preparedness and response [1,2,3]. There are currently two key UN initiatives, the first is the 2030 Agenda with 17 sustainable development goals, accompanied by global indicators. The second is the Sendai Framework dedicated to Disaster Risk Reduction (DDR) [4], with seven main goals and global indicators. China is active in both initiatives and comes up with solutions. The scale of its natural disasters makes China one of the “laboratories” where other countries and their experts learn how to understand and manage such events. For example, results from expert analyses have shown that there is a statistically profound association between overall influenza activity and influenza A activity and average relative humidity.

Another example comes from the novel coronavirus pneumonia epidemic (“COVID-19 epidemic” for short), which is a once-in-a-century infectious disease in human society. Its wide, rapid spread and difficult prevention and control are unprecedented since the founding of China. As of 24:00 on 12 December 2022, there are 249,168 asymptomatic cases of COVID-19 in China, including 30,879 deaths and a 12.3% fatality rate [5]. In the context of COVID-19, tourism, catering, entertainment, transportation, and other industries have been severely impacted. Enterprises have stopped production entirely or in part, schools have closed, and communities have been closed. The economy and society have been affected in many ways. In the face of the “Black Swan” incident of COVID-19, the Level I response to public health emergencies has been launched in the country. Local governments at all levels quickly launched emergency management efforts. These governments actively played a role in early warning monitoring, emergency response, rehabilitation, and other aspects of the ability to respond to public crises.

The Chinese government has not only made unrelenting efforts to deal with COVID-19 but has also conducted a lot of work in other emergency situations. The most important achievement is the formation of a mature system of “one case, three systems” in public emergencies [6]. On the basis of the system of “one case, three systems”, the Third Plenary Session of the 16th Central Committee of the CPC began to formulate a series of laws and regulations, such as the General Emergency Plan for National Public Emergencies and the Emergency Response Law of the People’s Republic of China. At the Third Plenary Session of the 19th Central Committee held in 2018, the emergency management function was reformed from the national department level, and the Emergency Management Department was formally established, forming a national security concept to deal with great emergencies. Since then, the modernization of the national governance system and its capacity has been promoted and developed, and the socialist system with Chinese characteristics has been further improved.

When each subject takes the maintenance of their own interests as the principle of action in the process of crisis response, the institutionalized cooperation mechanism is lacking and the response enters a state of disorderly competition, resulting in the failure of the entire emergency governance mechanism. The governance dilemma of emergency responses to major public health emergencies is mainly reflected in the imbalance between the demands for governance under a suddenly unconventional state and the fragmented supply of governance under the conventional state [7,8]. There is no doubt that various emergencies have brought severe governance challenges to governments at all levels. Local governments play a key role in emergency management. Their emergency management capability directly determines whether the harm and loss caused by COVID-19 in the region can be minimized, and also affects the further construction of government authority and credibility. Therefore, emergency response capacity building has once again become the focus of social concern and an important opportunity for local governments to strengthen local governance. At the same time, in the practice of government response to emergency events, it is of positive significance to explore the specific performance, achievements, governance logic, shortcomings, and causes of its emergency management capacity building. In addition, the experience of strengthening emergency response capacity building should be summarized, the management of public emergencies should be further improved, and the emergency response capacity of local governments should be enhanced.

## 2. Literature Review

The current academic research on the construction of the emergency management capacity of local governments mainly focuses on theoretical research on emergency response capacity and research on assessment systems of emergency management capacity.

### 2.1. Relevant Theories of Emergency Management Response

At present, there are few research results from Chinese and international scholars on the theoretical construction of government emergency management capability, and most of the mature theoretical research focuses on macro emergency management or crisis management. The “F model” proposed by Steven Fink, a crisis management expert; the “4R” theory advocated by Robert Heath, a management scholar; and the simple three-stage division model of “before, during and after the crisis event” clearly divide the crisis development and emergency management stages, providing a theoretical framework for the management of public emergencies.

To some extent, these mature theories have become important guides for the construction of the current government’s emergency management capacity. For example, based on the undeniable, objective fact of a risk society, the government should strengthen its in-depth thinking and discussion of emergency response capacity [9] and use the “4R” theory to carry out a specific analysis of its emergency management capacity [10,11,12]. On the basis of fully understanding the connotation of government emergency management, the government’s emergency management capability should cover all emergency management stages, including preparedness, response, disaster reduction, and recovery capabilities [13]. It is a kind of ability that should be possessed when dealing with emergencies and eliminating emergency dangers. This includes the ability to acquire and allocate resources, provide early warnings and rapid response, coordinate, disclose scientific research and information, handle crises or emergencies and their aftermath, etc. [14]. Emergency management is complex and includes multiple elements such as nature and society, hardware and software, manpower and systems, engineering ability, and organizational ability, which mainly reflects the comprehensive development ability of modern society [15]. Moreover, emergency management has a multi-level structure. At the national level, it is shown as a national governance capability with 13 key indicators, often used to deal with major crisis situations. Moreover, emergency management personnel should have certain professional qualities, including knowledge, scientific literacy, and leadership [16,17].

It is worth mentioning that in order to better deal with the challenges posed by disaster preparedness, crisis reduction, and post-disaster management, China and the European Commission signed the China–EU High-level Project on Disaster Risk Management (2012–2017) in 2012. In this project, China will learn from Europe on how to strengthen the national disaster crisis management system, and China and the EU will benefit from further dialogue and cooperation in this field. In addition, China and the EU will conduct regular exchanges in the areas of real-time capacity-building and science and technology, including early warning and forecasting tools [18].

### 2.2. Assessment System of Emergency Management Response

The assessment or evaluation object of emergency management capability includes the emergency management capability of a certain type of institution, a certain crisis event and even a certain city or region. The assessment of the government’s emergency management capability is often carried out around the main emergency management processes and stages, including the comprehensive use of expert scoring, key element verification, performance assessment, fuzzy comprehensive assessment and other methods. Research results include both model construction based on quantitative indicators and subjective judgment based on qualitative analysis.

In the 1990s, a set of CAR procedures for assessing emergency management preparedness was launched in the United States. This procedure systematically constructed 1014 specific indicators based on 13 functions such as disaster management, communication and early warning, and material management and analyzed the emergency preparedness of all 56 states, localities, and islands of the United States [19]. In 2002, the Fire Protection Agency of Japan, together with the Institute of Disaster Prevention and Information, jointly developed an assessment project specifically aimed at the disaster prevention capability of local public groups. They established a specific analysis system based on nine major projects to objectively measure the emergency management capability of public groups with figures as much as possible [20].

Chinese scholars also try to establish an assessment indicator system for emergency management capability and scientifically assess the government’s emergency management capability by establishing multi-level indicators and assigning corresponding weights. In terms of public health emergency management, it is typical to take the response preparation period, the early warning and monitoring stage, and the response process and aftermath stage as the first-level indicators; on this basis, the upper level indicators were refined and deconstructed, forming 20 s-level indicators, such as emergency plans, emergency teams, emergency supplies, and 48 third-level indicators, thus building a targeted and practical indicator system for assessing emergency response capacity [21]. From the dimensions of manpower, capital, material, information, technology, and group health, 12 first-level indicators and 56 s-level indicators were selected as an important basis for assessing the emergency response capacity of rural public health emergencies [22]. The government’s emergency management capability maturity is divided into five levels according to the actual needs of emergency activities, namely, “continuous improvement, quantitative management, defined, repeatable and initial levels”, and the characteristics of each level are clearly defined [23]. Moreover, the indicator system of the government’s early warning assessment should not only refer to the core elements of the capacity of each stage of emergency management process, but should also consider the final effect of emergency management and conduct a comprehensive analysis from dynamic and static aspects [24,25].

### 2.3. Summary Review

The above literature review systematically combs through the theoretical guidance, indicator models, or logical frameworks of government emergency management research and discusses the path to improving emergency management from the three levels of emergency management theory, assessment, and construction. The literature thus far has promoted the systematic development of government emergency management to a certain extent. However, when digging deep into its theoretical breadth and the dynamic nature of the used indicators, the following problems are prominent:

(1) In terms of research theory, there are many studies from the macro perspective, but the basic theoretical research needs to be deepened. At present, the research results on government emergency management can better explain the handling capacity of the Chinese government and others from the macro level. However, the biggest limitation is that the induction process and induction theory of the Chinese government’s emergency management events cannot be effectively explained, especially because of the lack of in-depth analysis of its dynamic assessment and control research from different levels, such as the situation assessment and prediction of emergency command, the preparation and release of emergency information, and cooperation and coordination preparation and assessment.

(2) In terms of research content, most research indicators are set up at the hardware conditions and static environment levels, and the assessment of dynamic and static elements is still a difficult point that current research is attempting to break through. For example, scholars have constructed many indicator models or logical frameworks which provide some reference for assessing the government’s emergency management ability. However, they have not considered the comprehensive and dynamic feedback and tracking of assessment indicators and also lack specific implementation measures, which has limited the practical effects of promoting the development of the government’s emergency management ability.

To sum up, to finally identify the application technology and assessment method system that can effectively and comprehensively guide the government’s emergency management practice, more innovative or detailed improvement work may be required. Under the background of the increasing attention being paid to government emergency management in China, it is of innovative significance to launch an assessment of the government’s emergency response in China from the basic perspective of the GB/T37228-2018 standard (Societal security—Emergency management—Requirements for incident response), which explains the theoretical basis of emergency management and detailed implementation measures. Therefore, this study selected a municipal government in Guangdong Province as the research case object and built its assessment system based on GB/T37228-2018 theory. From the basic perspective, we selected the two-tuple linguistic information method, which has greater advantages in language information processing, to assess the government’s emergency response level, with a view to improve its emergency response capability and further improve its emergency management status. In addition, this study also has certain reference significance for the establishment and improvement of other government emergency response systems.

## 3. Research Methodology and Data

### 3.1. GB/T37228-2018 Connotation

GB/T37228-2018 is equivalent to the international standard ISO22320:2011, namely, Societal security—Emergency management—Requirements for incident response. This standard can help public and private accident response organizations improve their ability to deal with various emergencies (e.g., crisis, destruction, and disasters) and realize the sharing of multi-level accident response functions among different organizations and institutions. The standard enables different organizations to carry out joint operations with minimum demand and maximum efficiency [26].

This international standard specifies the minimum requirements for effective accident response and provides a basis for factors such as command and control, business information, and the coordination and cooperation of an accident response organization. It includes the organizational structure and procedures of command and control, decision support, tracking, information management, and collaboration. It also establishes the demand for operational information of accident response, which is defined as timely, relevant, and accurate information generated through procedures, work systems, data collection, and management. In addition, it supports the process of command and control, as well as coordination and cooperation, within the organization and with other interested parties and defines the requirements for coordination and cooperation between organizations. The general framework of the GB/T37228-2018 standard is as follows [26]:

(1) Overview of the emergency command system.

The objective of establishing an emergency command system is to enable an organization to implement effective emergency response measures independently or with other participants to minimize casualties and property losses. In order to achieve emergency response objectives, the organization shall establish an emergency command system that complies with relevant laws, regulations, emergency plans, and the requirements of this standard. The emergency command process shall not be limited to the decision of the commander, but shall also include the decisions of the personnel at all levels whom assume command responsibility in the emergency headquarters. See Figure 1 for an example of the emergency command process of a emergency response organization in the case of a single command level.

(2) Overview of emergency information requirements.

In the process of emergency response, emergency information helps to effectively manage emergency response activities, perceive events, organize resources, and conduct emergency command. Emergency information includes information related to handling emergencies and response activities (see Figure 2). It can be information generated dynamically by events or static information related to locations, such as buildings, infrastructure, and population.

(3) Overview of cooperation and coordination requirements.

In order to achieve an effective emergency response, in the process of emergency preparedness, necessary cooperation agreements or plans should be formulated to support the cooperation between governmental organizations, non-governmental organizations, and international governmental organizations and non-governmental organizations. The above cooperation shall be based on risk identification and consequence assessment. The organization shall, on the basis of a necessity assessment, establish necessary coordination relationships with interested parties as part of emergency preparedness. The above command and coordination shall be based on risk identification and consequence assessment. See Figure 3 for the multi-level emergency command process.

### 3.2. Construction of Assessment Indicator System

According to the theories behind GB/T37228-2018, namely emergency management theory, crisis life cycle theory, capability maturity model theory, and the theoretical knowledge system of the United Nations Global Geospatial Information Management (GGIM), and after integrating papers [27,28,29,30] and other documents on emergency response assessment methods and strategies, it is proposed to build an assessment indicator system for government emergency response capability as shown in Table 1.

### 3.3. Tuple Linguistic Information

Due to the complexity of assessment indicators and the fuzziness and uncertainty of human thinking, it is most convenient and practical for assessors to give preference information in language when judging the indicators. However, previous methods used to convert indicator values of different degrees when dealing with linguistic information will cause certain information loss and distortion in the process of conversion, thus affecting the accuracy of the results [28]. In order to solve the problem of information loss caused by language information operation or processing, Professor Herrera of Spain put forward a method to describe language assessment information using two-tuple linguistic in 2000. This method uses the two-tuple form of a phrase and a real value in a predetermined set of language phrases to express all the information obtained after the integration of language information, which can effectively avoid the information loss and distortion in the aggregation and operation of language information, and is obviously superior to other language information processing methods in terms of calculation accuracy and reliability [28,31,32].

The two-tuple linguistic method is used to describe language assessment information. This method converts the preference information given by the decision maker into a two-tuple linguistic symbol $\left(s_{i},a_{i}\right)$ and then the decision analysis is carried out through some corresponding operations on two-tuple symbols, where $s_{i}$ is the i element in the predefined ordered natural language assessment set S composed of odd elements. In addition,$S=\left(s_{1},s_{2},\ldots s_{g}\right),s_{i}\in S$ is a language phrase. In this study, S is composed of five phrases, namely $S=\left(s_{1},s_{2},s_{3},s_{4},s_{5}\right)$, where $s_{1}=Initial\;Level$, $s_{2}=Preparation\;level$, $s_{3}=Process\;level$, $s_{4}=Management\;level$, and $s_{5}=Optimization\;level$. The classification of the grade and characteristics is shown in Table 2 (See the scoring criteria in Appendix A).

$a_{i}$ means the difference between the assessment result obtained after the assessment information given by the decision maker, which is aggregated by some algorithm and the closest language phrase $s_{i}$ in the initial language assessment set. a is a numerical value in the interval $\left[-0.5,0.5\right)$, and its definition of an aggregation operator can be expressed as [28,31,32]:

(1) Language phrase $s_{i}$ uses the following conversion function *θ* transformed into two-tuple linguistic form:$$\theta :S\to S\times \left[-0.5,0.5\right),\theta \left(s_{i}\right)=\left(s_{i},0\right),s_{i}\in S$$

(2) Suppose that real number $\beta \in \left[0,g\right]$ is the result of language information through some aggregation operator; its corresponding two-tuple linguistic form can be obtained by function Δ:$$\Delta :\left[0,g\right]\to S\times \left[-0.5,0.5\right),\Delta \left(\beta \right)=\left(s_{i},a_{i}\right)$$
where $i=round\left(\beta \right),a_{i}=\beta -1,a_{i}\in \left[-0.5,0.5\right)$, where *round* indicates the rounding operation.

(3) Correspondingly, if $\left(s_{i},a_{i}\right)$ is a two-tuple linguistic information, then there is an inverse function $\Delta^{-1}$, so that the two-tuple linguistic can be converted into the corresponding numerical value $\beta \in \left[0,g\right]$:$$\Delta^{-1}:S\times \left[-0.5,0.5\right)\to \left[0,g\right],\Delta^{-1}\left(s_{i},a_{i}\right)=i+a_{i}=\beta$$

(4) Suppose that $\left(s_{i},a_{i}\right)$ and $\left(s_{j},a_{j}\right)$ are any two-tuple linguistic information; the comparison of two-tuple linguistic is therefore orderly:➢$when\;i>j,\;\left(s_{i},a_{i}\right)>\left(s_{j},a_{j}\right),\;namely,\;\left(s_{i},a_{i}\right)\;is\;superior\;to\;\left(s_{j},a_{j}\right).$➢$when\;i=j,\;\mathrm{if}\;a_{i}>a_{j},\;\mathrm{then}\;\left(s_{i},a_{i}\right)>\left(s_{j},a_{j}\right),\;\mathrm{namely},\;\left(s_{i},a_{i}\right)\;\mathrm{is}\;\mathrm{superior}\;\mathrm{to}\;\left(s_{j},a_{j}\right).$



$\mathrm{if}\;a_{i}=a_{j},\;\mathrm{then}\;\left(s_{i},a_{i}\right)=\left(s_{j},a_{j}\right),\;\mathrm{namely},\;\left(s_{i},a_{i}\right)\;\mathrm{is}\;\mathrm{equal}\;\mathrm{to}\;\left(s_{j},a_{j}\right).$





$\mathrm{if}\;a_{i}<a_{j},\;\mathrm{then}\;\left(s_{i},a_{i}\right)<\left(s_{j},a_{j}\right),\;\mathrm{namely},\;\left(s_{i},a_{i}\right)\;\mathrm{is}\;\mathrm{inferior}\;\mathrm{to}\;\left(s_{j},a_{j}\right).$



(5) Suppose that $\left\{\left(s_{1},a_{1}\right),\left(s_{2},a_{2}\right),\ldots ,s_{m},a_{m}\right\}$ represents a group of two-tuple linguistic information, the arithmetic mean operator of this group of two-tuple linguistic information can be defined as: $$\left(\overset{ˉ}{s},\overset{ˉ}{a}\right)=\Phi \left(\left(s_{1},a_{1}\right),\left(s_{2},a_{2}\right),\ldots \left(s_{m},a_{m}\right)\right)=\Delta \left(\frac{1}{m}\sum_{i=1}^{m}\Delta^{-1}\left(s_{i},a_{i}\right)\right),\overset{ˉ}{s}\in S,\overset{ˉ}{a}\in \left[-0.5,0.5\right)$$

### 3.4. Data Acquisition

In order to further carry out the research, the government of H City in Guangdong Province of China was selected as a case for emergency response assessment. H City is located in the south of the China mainland, in the middle and lower reaches of the Han River, and is a port city in the east coast of Guangdong Province. It borders Zhao’an County and Pinghe County of Fujian Province in the east, Jiedong District of Jieyang City of Guangdong Province in the west, Fengshun County and Dapu County of Meizhou City in the north, and Shantou City and Chenghai District of Shantou City in the south. The land area of the city is 3146 square kilometers, and the sea area is 533 square kilometers. H City has a subtropical marine monsoon climate, with a mild climate and abundant rainfall. It is suitable for farming all year round. Due to the limitation in terrain, most of H City is built in valleys. The precipitation in these areas is relatively concentrated, and rainstorms are the main reason for this. Therefore, when the rainy season comes, geological disasters such as landslides, collapses, and debris flows often occur, and a series of problems such as shortages of materials are caused, posing a great threat to H City. Therefore, it is necessary to assess and improve the government emergency response. According to the design in Table 1 and Table 2, 120 relevant personnel from the Prevention and Control Office of H Government, the People’s Hospital, the high-speed railway station, the airport, H universities, enterprises, and other relevant personnel were invited over the past two years to complete questionnaire surveys (see Appendix A) and participate in online and offline interviews. After removing invalid questionnaires, 105 were valid. In order to ensure that each assessment indicator contributes to the system, this study uses SPSS 28 software to test the reliability and validity of the questionnaire. The result is that the value of Cronbach’s α is 0.920, which indicates that the questionnaire designed in this study is highly reliable. After the validity verification using KMO and Bartlett tests, it is concluded that the KMO > 0.8 and the coefficient of significance is less than 0.001, which indicates that the variables are effective and the correlation between variables is significant, so further modeling analysis can be conducted.

## 4. Result

Taking U_11_ as an example, according to the definition of two-tuple linguistic, the aggregated value of U_11_ can be calculated as: $$(\overset{ˉ}{s},\overset{ˉ}{a})=\Phi ((s_{1},a_{1}),(s_{2},a_{2}),\ldots (s_{102},a_{102}))=\Delta \left(\frac{1}{102}\sum_{i=1}^{102}\Delta^{-1}\left(s_{i},a_{i}\right)\right)=2.75=\left(s_{3},-0.25\right)$$

Similarly, the two-tuple linguistic information of other indicators can be obtained as shown in Table 3.

According to the definitions, the two-tuple linguistic information value of U_1_ can be calculated as:$$\begin{array}{ll} \left({\overset{ˉ}{s}}_{1},{\overset{ˉ}{a}}_{1}\right) & =\Phi ((s_{11},a_{11}),(s_{12},a_{12}),(s_{13},a_{13})(s_{14},a_{14}),(s_{15},a_{15}),(s_{16},a_{16}))\\ & =\Delta (\frac{1}{6}\sum_{i=1}^{6}\Delta^{-1}\left(s_{i},a_{i}\right))\\ & =\Delta \left(\begin{array}{l} \left(0.167\times 2.75\right)+\left(0.167\times 2.67\right)+\left(0.167\times 2.59\right)+\left(0.167\times 4.00\right)\\ +\left(0.167\times 2.68\right)+\left(0.167\times 2.83\right) \end{array}\right)\\ & =2.93\\ & =\left(s_{3},-0.07\right) \end{array}$$

Similarly, the two-tuple linguistic information of other indicators can be obtained as shown in Table 4.

According to the calculation results in Table 4, the overall level of government emergency response assessment from the perspective of the GB/T37228-2018 standard can be calculated as:$$\begin{array}{ll} (\overset{ˉ}{s},\overset{ˉ}{a})= & \Phi ((s_{1},a_{1}),(s_{2},a_{2}),(s_{3},a_{3})(s_{4},a_{4}),(s_{5},a_{5}),(s_{6},a_{6}))\\ & =\Delta (\frac{1}{6}\sum_{i=1}^{6}\Delta^{-1}\left(s_{i},a_{i}\right))\\ & =\Delta \left(\begin{array}{l} \left(0.167\times 2.93\right)+\left(0.167\times 2.94\right)+\left(0.167\times 2.95\right)+\left(0.167\times 3.17\right)\\ +\left(0.167\times 2.34\right)+\left(0.167\times 2.71\right) \end{array}\right)\\ & =2.86\\ & =\left(s_{3},-0.14\right) \end{array}$$

The final assessment results of this study show that the membership degrees of U_23_ (Situation assessment and prediction), U_27_ (Feedback and tracking), U_45_ (Emergency information assessment and feedback), U_54_ (Collaborative situation assessment and prediction), U_61_ (Implement information sharing and situation assessment strategies), and U_69_ (Ensure continuity of coordination process) belong to the Preparation Level; the membership degrees of U_14_ (Emergency resource management), U_33_ (Propose the method and result requirements for information collection), U_41_ (Emergency information collection), U_63_ (Decompose and assign emergency response tasks), U_64_ (Formulate and implement emergency support plan) belong to the Management Level; the membership degrees of other secondary indicators belong to the Process Level; the membership degree of U_5_ (Cooperation and coordination preparation) belongs to the Management Level; the membership degrees of other primary indicators are Process Level; and the membership relationship of the overall assessment result is Process Level.

In conclusion, H Government has established a basic command process, emergency information process, and cooperation and coordination process for emergency response. The management of emergency resources, information collection methods, response and guarantee plans, and other aspects are relatively standardized, which is worth promoting to the preparation and process management systems of other governments. The H Government’s disadvantages are the lack of a systematic improvement process and the imperfect preparation in situation assessment and prediction, information sharing and feedback, coordination process continuity, etc., which requires strengthening process management and tracking.

## 5. Deep Level Mechanism Analysis and Enlightenment

Based on the above assessment results, it can be observed that H Government has made outstanding efforts in the early stage of emergency management, such as emergency resource management, but is not perfect in the middle and late stages of emergency management, which is mainly reflected in three aspects: situation assessment, information sharing and feedback, and the continuity of the coordination process. In order to continuously improve the government’s emergency management capability, this study further analyses its three weak links.

### 5.1. Situation Assessment

Spatial awareness is to think of space in a transcendental intuitive cognitive way; space becomes a part of consciousness and is interpreted as a cognitive and experiential process of consciousness [33]. Situation assessment is not only cognition and experience, but also the process of prediction. The functional description of situation assessment is the process of explaining the status of a battlefield and identifying the enemy’s intention and operational plan according to the force deployment, operational capability, and effectiveness of all participating parties. Situation assessment is a multi-layer view to establish the organizational form of elements such as operational activities, events, time, locations, and force. It explains and represents the battlefield situation, points out the enemy’s behavior mode, infers the enemy’s intention, and forecasts changes in the future situation based on the mutual analysis of situation elements [34]. Endsley believed that the understanding of information is a process in which elements of the current situation in the time and space domains are detected, recognized, understood, and predicted [34]. Situation assessment is further interpreted as decision makers’ modes of thinking about the current situation. Therefore, situation assessment is divided into three levels, as shown in Figure 4.

It can be seen from Figure 4 that situation assessment can be understood as a pattern recognition problem with multiple levels, targets and objects. To solve the problem, it is necessary to gradually reach the optimal result according to historical data and constantly updated data. At the same time, it is necessary to establish situation descriptions from different views according to the needs of users. Therefore, in situation assessment, the government should fully understand the importance of big data technology in the process of emergency response governance. This is not only reflected in improving the importance of big data technology from the perspective of top-level design, but also in promoting the capacity building of emergency management with the support of big data technology. With the help of big data technology, obtaining statistics and conducting analyses of urban risk factors in a timely manner can enhance urban risk prevention. The main goal of risk prevention and of control when risks occur should be based on daily data and the use big data technology to conduct information mining, provide a strong basis for scientific prediction, and take effective measures to prevent risks from occurring in a timely manner.

### 5.2. Information Sharing and Feedback

Information sharing and feedback is to control risks according to the information of system output changes, that is, to obtain the expected system’s performance by comparing the deviation between system behavior (output) and its expected behavior and eliminating the deviation. The main features of the information sharing and feedback dynamic decision-making process include the following, as shown in Figure 5 [35]:(1)The information sharing and feedback decision-making process is not a passive response, but an active feedback adjustment.(2)The information sharing strategy based on feedback is a process in which the system continuously adjusts and changes the strategy until it reaches a goal.(3)The process of information sharing and feedback decision making is a process of learning, evolution, and adaptation which is constantly undergoing dynamic adjustment with the evolution of the environment.(4)The dynamic distribution of information sharing and feedback needs the support of information networks. In the process of information sharing and feedback operation, the feedback information must be timely and accurate; otherwise, no decision can be made or wrong decisions can be made.

According to the four characteristics above, after the disaster emergency alarm is issued, the decision-making body should collect relevant intelligence information in a timely manner and draw up a scientific, reasonable, and feasible emergency support plan based on their judgment. On the basis of this effective information and on the comprehensive research and analysis of the emergency management organization on the nature and intensity of the disaster event, the decision-making body should cooperate to implement the decision. Through an efficient emergency information system, the decision-making body should track and supervise the emergency management operation without interruption, and collect and sort out feedback information in a timely manner. If it is found that there are loopholes in the supply of emergency decision-making demand, the decision-making body shall coordinate with the cooperative body to take corresponding measures to adjust or correct the deviation between the decision-making process and the plan, so as to achieve an adaptive effect. The above steps should be repeated to guide the emergency decision making until the needs of the emergency are satisfied, which is also the continuity and core idea of the coordination process. The dynamic decision making and coordination continuity diagram is shown in Figure 6.

## 6. Limitations

Although this study has proven that the assessment of government emergency management based on the GB/3722-2018 standard is feasible and comprehensive, there are still some deficiencies. First, while the assessment indicators constructed can fully reflect the terms of the GB/3722-2018 standard, the awareness of emergency management of relevant personnel, emergency culture construction, and other issues are not reflected, which may require continuous application and improvement of the GB/3722-2018 standard in practice. Second, due to the complexity of the assessment indicators and the fuzziness and uncertainty of human thinking, it is better for assessors to give preference information in linguistic form when judging assessment indicators; however, two-tuple linguistic information is only a semi quantitative assessment method, and it is difficult to obtain satisfactory results completely and objectively. Third, due to the dynamic complexity of assessment indicators, the government also needs to dynamically assess the results of emergency management, therefore, with the integration of time, place, and other variables, the assessment process needs more dynamic management. Four, previous studies have shown that there was a statistically profound association between overall influenza activity and influenza A activity for average relative humidity [36]. In all aspects of emergency response, geospatial data and tools may help to save lives, limit damage, and reduce the cost of social response to emergencies. Therefore, in order to comprehensively evaluate the government’s emergency response, geospatial data and tools should be integrated into the response and recovery of emergency management, and then into the mitigation of future events [37]. Therefore, it is necessary to fully consider the relevant information of geographical space (such as temperature and humidity) for the design of evaluation indicators. It is worth mentioning that its assessment results and improvement measures and the data related to space information exchange and sharing need to be updated in GGIM in time.

## 7. Conclusions

This study looks at the government emergency response from the perspective of the GB/T37228-2018 standard as the entry point, conducts a case analysis on the construction of emergency management capacity of H Government; determines the advantages, governance logic, and shortcomings of its assessment results; and puts forward suggestions to promote the optimization of local government emergency management capacity.

(1) Aiming at the long process of emergency resource management, the H Government should continue to standardize these processes in emergency resource management, information collection methods, etc., and promote them to other emergency response processes, so that the preparation stage and process stage of emergency response can be continuously optimized and improved in practice and the government’s emergency response ability can be improved. In general, in emergency response, the work in the early stages of the response guarantees or sets the precedent for the later management of emergency information and cooperation and coordination. The continuous standardization and improvement of the whole emergency response process is the key to improving the government’s emergency response capacity.

(2) Aiming at the short process of situation assessment, the uncertainty of an emergency is an important factor that causes the greatest harm. An emergency’s uncertainty is mainly manifested in the uncertainty of its cause, of the process from the impact to the result, and the final result. At the same time, different judgments about the results of events will also lead to different means of handling events. The subtle differences in each link will eventually lead to completely different results. It has been proven that some scientific methods, such as the application of big data technology, quantitative assessment and prediction, expert analysis, and mass interviews, are reliable and effective means to prevent and control emergencies. Therefore, strengthening situation assessment and prediction is an important measure for the government to deal with various uncertain events and reduce losses caused by uncertain events.

(3) Aiming at the short process of information sharing and feedback, a scientific information processing system often means minimizing the interference from value bias on information processing, thus strengthening the independent analysis of experts and scholars and absorbing the opinions of a wider range of social members. The use of such as system also means finding the rules in chaotic information, mainly including the improvement of laws and regulations relevant to information sharing and establishing a fair and open disaster record database. In addition, it is also important to design a standardized data system that can be used by all departments and organizations and formulate the system specifications for database updating and optimization. The local government where the emergency occurred is the first level of government that knows the situation best and responds first, but it also has the least materials, manpower, and power. The corresponding superior government has more resources, but the information it has is not as real and effective as that of the grassroots government. There is thus a complementary relationship between the two. Therefore, a smooth and efficient coordination mechanism should be established between the superiors and subordinates and between various government departments to achieve information sharing and resource sharing and maximize the rescue efficiency.

(4) Aiming at the short process of coordination process continuity, today’s emergencies cover a wider range. An event often involves multiple fields, becoming more unpredictable and complex to handle. Conventional means are often unable to deal with these emergencies, and resources outside the region need to be mobilized to deal with it. This COVID-19 epidemic involved many departments, such as the Health Commission, civil aviation, the railway, the Military Commission, the Development and Reform Commission, the Ministry of Finance, the Ministry of Education, the Ministry of Foreign Affairs, and so on. The coordination and cooperation of all parties and departments is thus required to minimize the impact of emergencies. Therefore, in the process of responding to public emergencies, different governments have different responsibilities due to regional and jurisdictional factors. It is necessary to strengthen the coordination ability among local grassroots governments so that they can cooperate and communicate with each other when dealing with emergencies. All localities need to establish an emergency response system focusing on “unified leadership, comprehensive coordination, classified management, hierarchical responsibility and territorial management”, including the main leaders of local governments at the grassroots level, emergency offices, relevant professional departments, and expert groups. The government should strengthen its own coordination ability so that it can comprehensively manage all kinds of emergencies in all aspects of emergency response and supervise the behavior of functional departments in the process of emergency response so that the coordination process is more capable of learning, evolving, and adapting, and constantly and dynamically adjusting with the evolution of the environment.

In short, the emergency response assessment system proposed in this study is a process-oriented assessment method which can better reflect the comprehensive management effect of the government in the process of emergency response and disposal. The established model can help local governments (or departments) find the shortcomings in their emergency response plans, identify the weak links in management, and establish targeted improvement plans so that their emergency response capacities can be gradually and continuously improved. The research results are of positive significance for exploring the improvement of government emergency response capability in complex environments and help to obtain comparable assessment conclusions through the assessment of emergency response processes, giving consideration to dynamic and static effects. The results can also be a guide to governments at all levels to establishing a continuous improvement mode of dynamic organization and scheduling capabilities, so as to improve the quality and effectiveness of their emergency management.

## 8. Directions for Future Research

This study assessed the government’s emergency response capability from the perspective of GB/37228-2018 standardization. The assessment is not limited to the specific application of GB/37228-2018 standardization theory, but it also broadens the assessment of government emergency response and strengthens the standardization and process of emergency management. However, it has to be said that the GB/37228-2018 standard’s terms lack some tacit knowledge required for emergency response, such as emergency concepts, risk awareness, psychological intervention mechanisms, people-oriented response, and other important concepts. Therefore, how to build implicit and explicit assessment indicators that comprehensively reflect the government’s emergency management ability is still a challenge. In addition, government emergency response has been developing continuously in the past two decades, but difficulties such as the inconvenience of data collection, the imprecision of qualitative or semi-quantitative assessment, and the dynamics of assessment indicators are still being faced. The time and space variables incorporated in the assessment of some measurement models also need further empirical research to ensure the effectiveness and systemization of the assessment.

## Figures and Tables

**Figure 1 ijerph-20-05091-f001:**
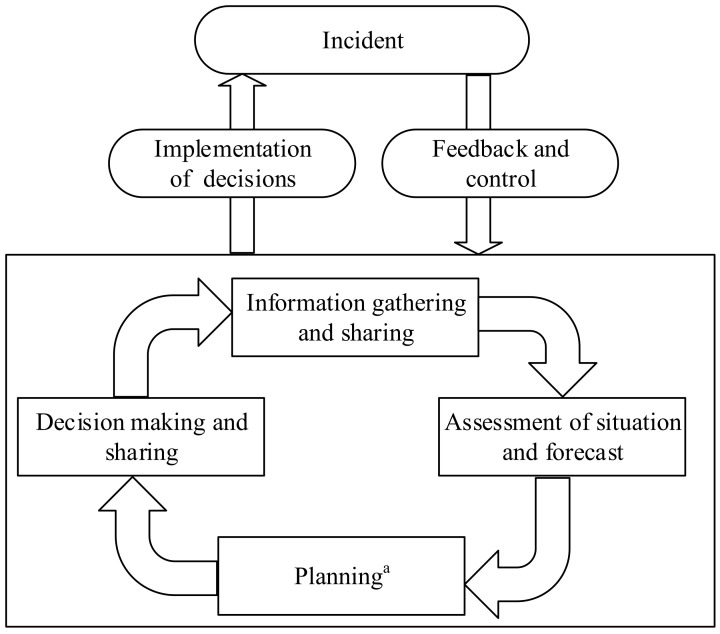
Example of command and control process in a single hierarchical organization with limited coordination needs. ^a^ with limited need for coordination with partners outside the organization.

**Figure 2 ijerph-20-05091-f002:**
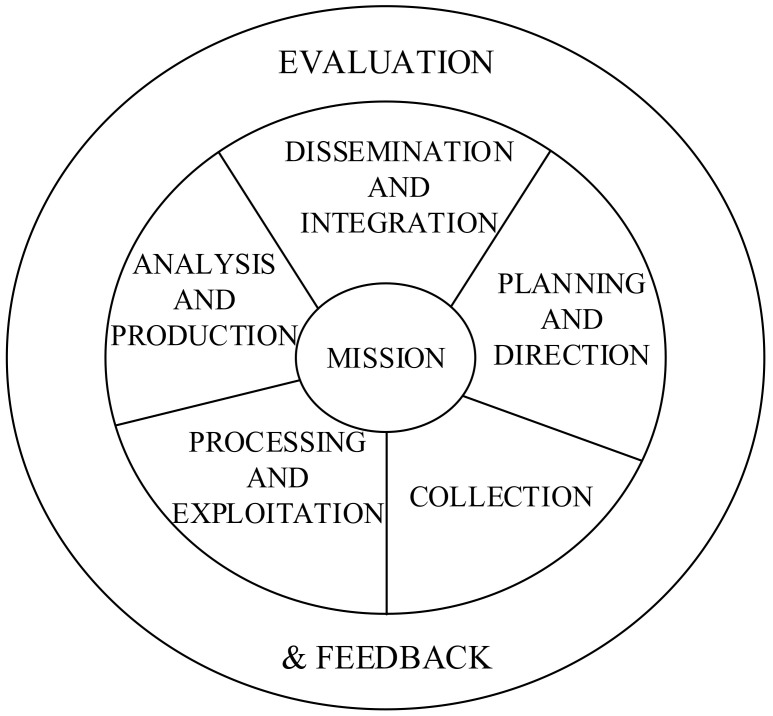
Process of providing operational information.

**Figure 3 ijerph-20-05091-f003:**
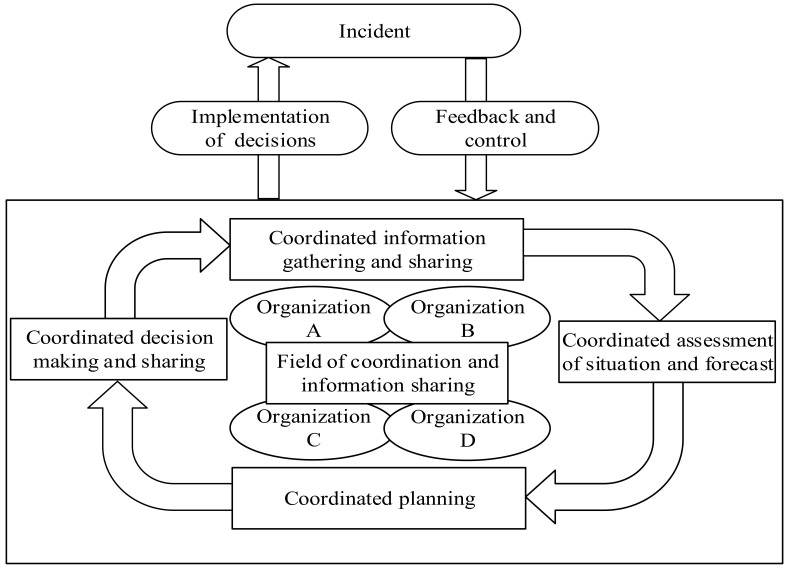
Circular chart for a multiple hierarchical command and control process with enhanced relevance of coordination.

**Figure 4 ijerph-20-05091-f004:**
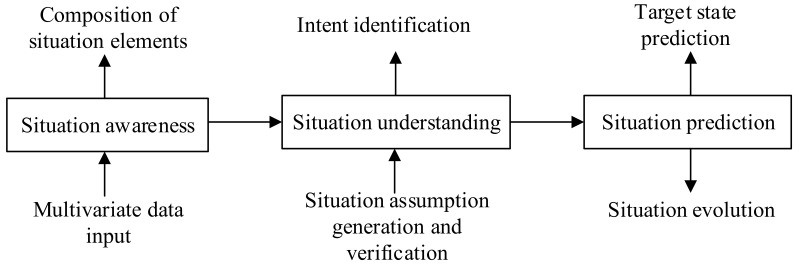
Functional model of situation assessment.

**Figure 5 ijerph-20-05091-f005:**
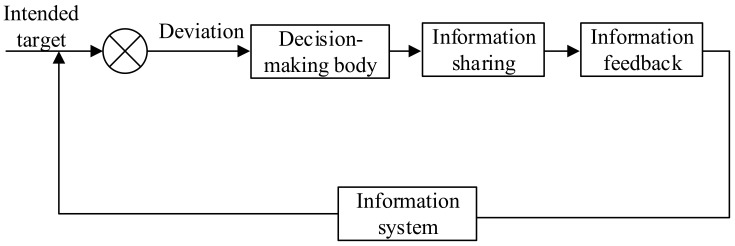
Information sharing and feedback system.

**Figure 6 ijerph-20-05091-f006:**
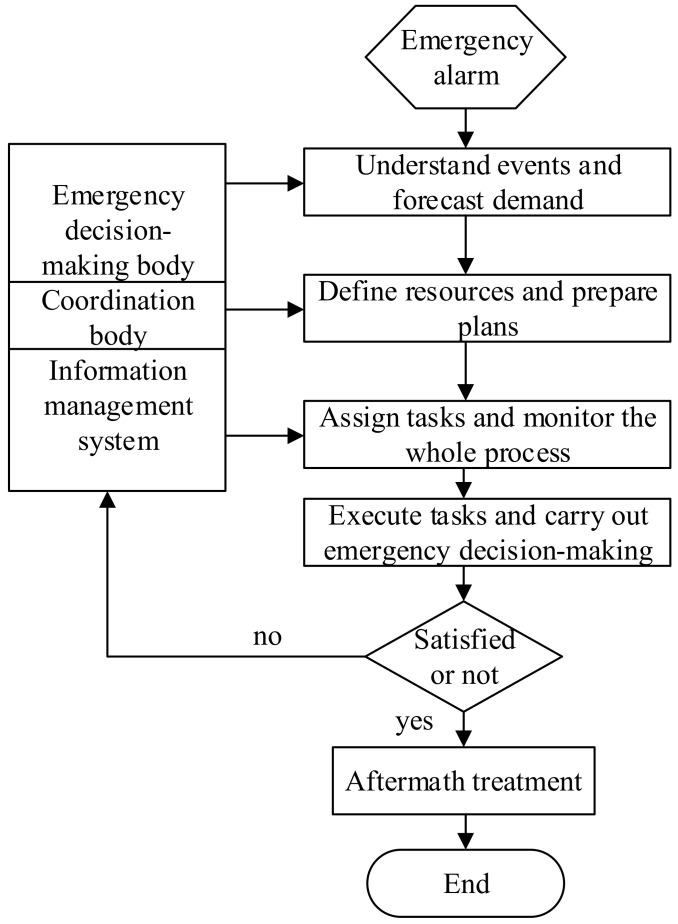
Dynamic decision making and coordination continuity flow chart.

**Table 1 ijerph-20-05091-t001:** Emergency response assessment indicator system inspired by GB/T37228-2018.

U_1_ Emergency command preparation	U_11_ Develop and update emergency response objectives
	U_12_ Determine the roles, responsibilities, and interrelationships
	U_13_ Formulate rules, restrictions, and plans for emergency response
	U_14_ Emergency resource management
	U_15_ Ensure compliance with relevant laws and regulations and responsibilities
	U_16_ Record key decisions and decision bases
U_2_ Emergency command process	U_21_ Emergency site investigation
	U_22_ Information collection, processing, and necessary sharing
	U_23_ Situation assessment and prediction
	U_24_ Develop emergency plan
	U_25_ Issue emergency plan
	U_26_ Implement emergency plan
	U_27_ Feedback and tracking
U_3_ Emergency information preparation	U_31_ Determine the objectives of response tasks and provide guidance
	U_32_ Explain the key issues for effective decision making
	U_33_ Propose the method and result requirements for information collection
	U_34_ Specify information storage, use, and access rights and restrictions
	U_35_ Stipulate information processing equipment and operation management
	U_36_ Determine time limits for information needs
	U_37_ Determine requirements and protocols for information dissemination
	U_38_ Determine the information needs of each participant
	U_39_ Specify the personnel related to emergency information
U_4_ Emergency information process	U_41_ Emergency information collection
	U_42_ Emergency information processing and use
	U_43_ Emergency information analysis and generation
	U_44_ Emergency information release and integration
	U_45_ Emergency information assessment and feedback
U_5_ Cooperation and coordination preparation	U_51_ Develop cooperation agreement or plan
	U_52_ Assess the command and coordination relationship with interested parties
	U_53_ Collaborative information collection, processing, and sharing
	U_54_ Collaborative situation assessment and prediction
	U_55_ Cooperate to formulate emergency plan
U_6_ Cooperation and coordination process	U_61_ Implement information sharing and situation assessment strategies
	U_62_ Determine information reporting and communication process
	U_63_ Decompose and assign emergency response tasks
	U_64_ Formulate and implement emergency support plan
	U_65_ Set boundaries between different organizations
	U_66_ Implement specific resource management
	U_67_ Ensure connectivity
	U_68_ Identify key requirements
	U_69_ Ensure continuity of coordination process

**Table 2 ijerph-20-05091-t002:** Grade and characteristics of government emergency response capability level inspired by GB/T37228-2018.

Grade	Characteristics
*s*_1_ Initial Level	The emergency management process is disordered or even chaotic. There are often insufficient emergency management command preparation, incomplete information collection, processing and sharing, and no corresponding situation assessment and prediction.
*s*_2_ Preparation level	Basic emergency management preparation has been established. Emergency command has been prepared, and corresponding command preparation, emergency information preparation, and cooperation and coordination preparation have been established. Other process preparations are incomplete.
*s*_3_ Process level	Basic emergency management process has been established. Corresponding command process, emergency information processing process, and cooperation and coordination processing process have been established. However, there is no systematic improvement process.
*s*_4_ Management level	The emergency command process has been standardized. Emergency command preparation and process, emergency information preparation and process, cooperation and coordination preparation, and process management are standardized.
*s*_5_ Optimization level	It has been able to continuously improve the process. Each preparation stage and process stage can be continuously optimized and improved in practice.

**Table 3 ijerph-20-05091-t003:** Summary of Two-Tuple Linguistic Information of two-level indicators.

Indicator	Two-Tuple Linguistic Information	Indicator	Two-Tuple Linguistic Information	Indicator	Two-Tuple Linguistic Information	Indicator	Two-Tuple Linguistic Information
U_11_	(s_3_,−0.25)	U_26_	(s_3_,−0.18)	U_41_	(s_4_,0.34)	U_62_	(s_3_,0.36)
U_12_	(s_3_,−0.33)	U_27_	(s_2_,0.28)	U_42_	(s_3_,0.27)	U_63_	(s_4_,−0.39)
U_13_	(s_3_,−0.41)	U_31_	(s_3_,0.34)	U_43_	(s_3_,−0.28)	U_64_	(s_4_,−0.27)
U_14_	(s_4_,0.00)	U_32_	(s_3_,0.22)	U_44_	(s_3_,−0.21)	U_65_	(s_3_,−0.22)
U_15_	(s_3_,−0.32)	U_33_	(s_4_,−0.25)	U_45_	(s_3_,−0.19)	U_66_	(s_3_,−0.46)
U_16_	(s_3_,−0.17)	U_34_	(s_3_,−0.28)	U_51_	(s_2_,−0.24)	U_67_	(s_3_,−0.32)
U_21_	(s_3_,0.12)	U_35_	(s_3_,−0.21)	U_52_	(s_3_,−0.37)	U_68_	(s_3_,−0.49)
U_22_	(s_3_,0.23)	U_36_	(s_3_,−0.41)	U_53_	(s_3_,−0.21)	U_69_	(s_2_,−0.30)
U_23_	(s_2_,0.26)	U_37_	(s_3_,−0.29)	U_54_	(s_2_,−0.28)		
U_24_	(s_3_,−0.33)	U_38_	(s_3_,−0.28)	U_55_	(s_3_,−0.18)		
U_25_	(s_3_,−0.21)	U_39_	(s_3_,−0.22)	U_61_	(s_2_,0.38)		

**Table 4 ijerph-20-05091-t004:** Summary of Two-Tuple Linguistic Information of one-level indicators.

Indicator	Two-Tuple Linguistic Information
U_1_	(s_3_,−0.07)
U_2_	(s_3_,−0.06)
U_3_	(s_3_,−0.05)
U_4_	(s_3_,0.17)
U_5_	(s_2_,0.34)
U_6_	(s_3_,−0.19)

## Data Availability

The data analysis of this study can be obtained from the corresponding author upon reasonable request.

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
