# Peer review of "Government Emergency Response Assessment from a Novel Perspective of GB/T37228-2018—A Case Study in China"

_ijerph, 2023, doi:10.3390/ijerph20065091_

Round 1

Reviewer 1 Report

1. It was not easy for me to read and understand the contents of this manuscript. All documents to include this article in the field of emergency management should be written easy, when reflecting that many emergencies occur within a short period of time in general. I hope that the authors will change the whole manuscript into easy structures and/or contents. 

2. As a similar token, many sentences in this manuscript were too long. Please make them various short sentences. (A few journals have asked writers to consist of at least three sentences in a paragraph.) Long sentences made me feel difficult to understand the manuscript. 

Reviewer 2 Report

The reviewer appreciates the authors for choosing the necessary topic, even if the authors themselves did not always fulfill their own critical tones and did not always go concretely into practical examples at the local level. The reviewer has several important comments and recommendations.

First of all, the reviewer wants to recommend the authors at least briefly include the issues of early warning (EW) and disaster risk management(DRM) in the international context. There are currently two key UN initiatives, the first is the 2030 Agenda with 17 sustainable development goals, accompanied by global indicators. The second is the Sendai Framework dedicated to Disaster Risk Reduction (DRR), also with 7 main goals and global indicators. In both initiatives, China is active and comes up with solutions. The very scale of natural disasters makes China one of the "laboratories" where other countries and their experts learn how to understand and manage such events.

In this context, the article also lacks a mention of DRR, as a broader concept dealing with the issue of disasters and preparing for them, even on a local scale. For an early warning, the conclusions of Hyogo (The Hyogo Framework for Action (2005-2015) (HFA): Building the Resilience of Nations and Communities to Disasters), developed after the earthquake and tsunami in South East Asia in late 2004, should be mentioned, which clearly (and how is also proving to be successful) improve the procedures used on an international scale.

The authors provided a professional overview of foreign and domestic works, yet some are surprisingly missing. For example, the high-level project EU-China Disaster Risk Management (2012-2017), was guaranteed on the Chinese side by the State Bureau and the University of Governance in Beijing, on the EU side it was the European Commission and the French Ministry of Interior. The project compared the situation in EU and China thoroughly, even on a global level. The authors do not mention it.

What the reviewer lacks the most is the absolute absence of understanding of the meaning of space and spatial (geographical) data and information when dealing with crisis situations. The UN initiatives mentioned above are also supported by the U.N. GGIM (Global Geospatial Information Management), which tries to integrate both existing and emerging spatial data infrastructure and create a Global Data Ecosystem. The data obtained and maintained in this way can then be used to solve EW, DRM, and DRR problems on global, regional, and local levels. Geospatial Databases are being created in a number of places in China. In addition, their cartographic interpretation is expected not only for experts but also for the general public.

It is right the authors several times talking about COVID-19, but just during it and earlier after Hurricane Katrina in the USA was evident the very important role geographic (geospatial) data and information.

In the first case, I would recommend authors to be inspired by the  book:

COVID-19 Pandemic, Geospatial Information, and Community Resilience.  Global Applications and Lessons. Edited By Abbas Rajabifard, Daniel Paez, Greg Foliente.1st Edition, 2021. Taylor Francis. Imprint CRC Press. 58 p. DOI https://doi.org/10.1201/9781003181590

In the second case, the NRC published post-Hurricane Katrina, which was a very cautionary example of how administration, even in a rich country, can fail completely. The book is actually an assessment by leading American experts of what went wrong and what needs to be prepared so that it goes well in the future. The overview of things that already exist and those that were missing, including the local level, is inspiring. Citation is here:

National Research Council. 2007. Successful Response Starts with a Map: Improving Geospatial Support for Disaster Management. Washington, DC: The National Academies Press. https://doi.org/10.17226/11793.

From a general point of view, the reviewer recommends a special double issue of the International Journal for Digital Earth (IJDE), where the philosophy of the approach to the solution and the characteristics of the current state are already given in the introductory texts:

Early warning and disaster management: the importance of geographic information (Part A) Konecny, M and Reinhardt, W., International Journal of Digital Earth, 3 (3) , pp.217-220. Taylor & Francis Ltd. 2010|.

Early warning and disaster management: the importance of geographic information (Part B), Konecny, M and Reinhardt, W., International Journal of Digital Earth, 3 (4) , pp.313-315.Taylor & Francis Ltd. 2010|.

What surprised me is the authors' ignorance or underestimation of China's progressive steps in the field of thematic maps and spatial data. It is mainly a CASM (Chinese AScademy for Surveying and Mapping) production dedicated to crisis situations or Covid-19.

To be understood correctly. I know about the professional orientation of the authors of the article, who are from other fields than those that work with spatial data and information. But its possible involvement, the manifestation of the awareness of the importance of this spatial aspect, and the opinion of the authors of the article, how they could further guide the implementation of the plans or improve could show people in Local government new available options for solving crisis situations.

On the other hand, I commend the authors for their great efforts to create a framework for early warning and crisis management solutions down to a truly local level.

Some recommendations.

I recommend authors reduce the text between lines 54 and 75 to a minimum. If the article was intended only for the Chinese community, I wouldn't object. But this professional scientific article is intended for the international community, which may or may not be oriented in various political aspects. I recommend mentioning what documents exist, giving a brief link to them, and listing them in References.

Reviewer recommends clarifying or defining relations between Spatial awareness and Situational assessment.

The authors use the term Big data technologies correctly, but there is a lack of specification of how the authors understand big data and which technologies they are specifically talking about.

If the reviewer suggests accepting the article with major changes, he means the ones he just described. They think they can be supplemented.

The article is written on the basis of an honest study of the situation, scientific erudition and international experience and knowledge, and is supplemented with comprehensible figures and tables. Clarification of this study uses is very valuable 2-tuple linguistic information method to assess the indicator system of H Government of China constructed according to the GB/T37228-2018 standard (Societal security-Emergency management-Requirements).

Critical comments on what needs to be improved are also in order, as well as warnings about challenges of the implicit knowledge of emergency management, the integration of time and space variables, and other issues (using the authors' own words). The authors may not have realized it, but their progression through headline targets and indicators is very similar to that developed by the U.N.

The results described in the article can be inspiring for other countries and the wider scientific and professional community

Reviewer 3 Report

This research used 2-tuple linguistic information method to assess the indicator system of H Government of China according to the GB/T37228-2018 standard.The research perspective is relatively new. However, I have some suggestions for the authors.

1. In the introduction section, there are too many descriptions on the general background. Please simplify these contents, so as to enter the topic more quickly. Considering the topic, this section should focus on the current situation about emergency response assessment, so relevant descriptions should be added. Furthermore, the scientific problems to be solved must be further clarified.

2. Whether the research target is emergency management or emergency response assessment? If they cannot be separated, please explain in detail. If not, please keep one term. I guess the emergency response assessment is more suitable.

3. What is the essential relationship between your research and reviewd literature? It is not the fields involved in your research that need to review, but the help or inspiration for you. Literature review decides what you should do on your topic. From this point of view, 2.1 and 2.3 are not necessary.

4. The webpage of refernce [35] cannot display GB/T37228-2018. Please provide a more reliable source.

5. Although the geography situation of H city is described in 3.4, the reasons why it is selected to be the case are not clarified. Generally, only if a city is vulnerable to damage is it necessary to study, so please introduce the risk and vulnerability of H city.

6. What are contents of the questionnaire? If available, please provide it in appendix or supplementary material. In addition, there is no need to repeat the tables and figures in supplementary material.

7. There is a weak correlation between the assement and conclusions.

8. I suggest the authors adopt more Internationalized expressions, e.g. "China mainland" is more appropriate than "motherland", "Chinese" is better than "domestic".

9. Some sentences contain too many contents, please split long sentences and improve language.

Round 2

Reviewer 3 Report

The revision is well-done, while I still have some suggestions.

1. Suggest the authors providing an official website for reference 26.

2. Suggest the authors putting the questionnaire in Appendix.

Author Response

I have provided the official website for reference 26 as shown below:

https://std.samr.gov.cn/gb/search/gbDetailed?id=7E2903B0D6B55A63E05397BE0A0AF660

Also, I have put the questionnaire in Appendix.
